# Fabrication of Conductive Fabrics Based on SWCNTs, MWCNTs and Graphene and Their Applications: A Review

**DOI:** 10.3390/polym14245376

**Published:** 2022-12-08

**Authors:** Fahad Alhashmi Alamer, Ghadah A. Almalki

**Affiliations:** Department of Physics, Faculty of Applied Science, Umm AL-Qura University, Al Taif Road, Makkah 24382, Saudi Arabia

**Keywords:** smart textile, SWCNTs, MWCNTs, graphene, applications

## Abstract

In recent years, the field of conductive fabrics has been challenged by the increasing popularity of these materials in the production of conductive, flexible and lightweight textiles, so-called smart textiles, which make our lives easier. These electronic textiles can be used in a wide range of human applications, from medical devices to consumer products. Recently, several scientific results on smart textiles have been published, focusing on the key factors that affect the performance of smart textiles, such as the type of substrate, the type of conductive materials, and the manufacturing method to use them in the appropriate application. Smart textiles have already been fabricated from various fabrics and different conductive materials, such as metallic nanoparticles, conductive polymers, and carbon-based materials. In this review, we study the fabrication of conductive fabrics based on carbon materials, especially carbon nanotubes and graphene, which represent a growing class of high-performance materials for conductive textiles and provide them with superior electrical, thermal, and mechanical properties. Therefore, this paper comprehensively describes conductive fabrics based on single-walled carbon nanotubes, multi-walled carbon nanotubes, and graphene. The fabrication process, physical properties, and their increasing importance in the field of electronic devices are discussed.

## 1. Introduction

Traditional textiles were created to protect people from the elements, such as cold and rain, and to serve as covering material. The two most important qualities associated with clothing are their ability to provide protection and their aesthetics. Throughout history, advances in smart materials and electronics have contributed to a unique potential that has led to the emergence of a new field called “smart textiles”. Smart textiles, also called intelligent textiles or e-textiles, are a type of intelligent materials that can detect and respond to changes in their environment [1]. The stimuli and responses can be thermal, electrical, magnetic, mechanical, chemical, or any other type of stimulus or response [2]. Smart textiles are indeed used in many applications ranging from simple to more complicated ones, for example, in military, healthcare, and wearable electronics [3,4,5]. They are classified into three groups based on their generation and intelligence [6,7]. First generation passive smart textiles can provide additional functions in a passive mode regardless of environmental changes. Examples of passive smart textiles include anti-odor, anti-static, anti-microbial, and bulletproof [8,9]. In the second generation, smart textiles have been developed to sense and respond to environmental stimuli. Examples include heat storage, sensors, thermoregulation, vapour-absorbing fabrics, and electrically heated suits [8]. A sophisticated smart textile consists primarily of an entity that functions similarly to the brain, with cognitive, reasoning, and activating capabilities that can sense, respond, and adapt to environmental conditions or stimuli, including health monitoring and space suits [10]. Figure 1 shows a chronology of the development of smart textiles. 

Although recent advances in the field of smart textiles are extremely interesting, several obstacles still need to be overcome to make them suitable for commercial and economic use [6]. Unfortunately, the fabrication of conductive textiles is limited by several technical and non-technical features. Therefore, it is essential to develop large-scale manufacturing processes [11,12]. One of the problems in developing smart textiles is the smooth and undetectable integration of the required electronics into the fabric. Therefore, material scientists need to develop fibers with the required electrical capabilities that are strong, comfortable, and attractive at the same time [13]. In addition, temperature, perspiration, humidity, mechanical shock, continuous bending and tension, and illumination should be thoroughly investigated [14]. The autonomy of the system should be increased to reduce the burden of frequent battery charging, and the battery life also needs to be improved, which is still a research problem [15]. The garments must ensure high security against cyber threats [16]. For users to fully embrace this new technology, smart clothing must be a product that meets consumers’ emotional and functional needs, and integration and connectivity tools [17].

This article is about the fabrication of smart textiles using carbon-based materials, especially SWCNTs, MWCNTs, and graphene, and is organized as follows: The smart materials, discussing the structure, physical properties, and potential applications of carbon nanotubes, SWCNTs and MWCNTs, and graphene. Then, there are three major sections focusing on the smart textiles fabricated with SWCNTs, MWCNTs and graphene, respectively. Meanwhile, the fabrication method, physical properties, especially electrical properties, factors affecting these properties, and potential applications of smart textiles are described.

## 2. Smart Materials 

Materials that are described as “smart” or “functional” are usually part of a “smart system” that can sense and respond to its environment. If they are truly intelligent, they have a significant impact on the performance of smart textiles [18]. In this article, the focus is on single-walled carbon nanotubes, multi-walled carbon nanotubes, and graphene.

### 2.1. Carbon Nanotubes (CNTs)

#### 2.1.1. Definition and Structure 

Carbon nanotubes (CNTs) belong to the fullerene family, which includes carbon allotropes whose atoms are connected in cage-like configurations, such as a hollow sphere, an ellipsoid, or a cylinder [19,20,21,22], and have a thickness or diameter on the order of a few nanometers [23]. CNTs can be fabricated in a variety of ways, but the three most common methods are fabrication by electric arc, chemical vapor deposition, and laser ablation [24].

#### 2.1.2. Types of Carbon Nanotubes

CNTs are generally classified by the number of carbon layers into single-walled (SWNTs) or multi-walled (MWNTs) carbon nanotubes, as shown in Figure 2. SWCNTs are single graphene layers wrapped in tubes. Depending on how the tube is wrapped, SWCNTs have different properties [25] and structures [26,27,28]. MWCNTs, on the other hand, consist of multiple graphite layers wound on top of each other [29], and the diameter between the tube walls is about 0.34 nm. The architecture of MWCNTs can be described by one of two models: the Russian doll model and the parchment model [30]. Table 1 shows the comparison between SWCNTs and MWCNTs and Table 2 shows the main physical properties of the two [31,32,33,34,35,36,37,38,39,40,41,42,43,44,45,46].

#### 2.1.3. Potential Applications of CNTs

CNTs were used in wide range of applications due to its small and lightweight, which makes them suitable for a [40]. They can be used in many fields, such as electronic and photovoltaic devices [47], solar cells [48], superconductors [49], food science [50], water purification [51], biology and medicine [52], electrical/electronic applications [53], wearable devices, and smart textiles [54]. Figure 3 shows various applications of CNTs in textiles.

### 2.2. Graphene

#### 2.2.1. Definition and Structure 

Graphene is a flat monolayer of carbon atoms densely condensed into a two-dimensional (2D) honeycomb crystal structure [55,56,57,58,59,60,61], as shown in Figure 4. Graphene belongs to the category of graphitic nanomaterials, which includes graphene with few layers (1–5 layers). It has numerous chemical [62,63], physical [64], electronic [65], and mechanical [66] excellent properties. In addition, graphene is said to be the thinnest known substance [67], the most hydrophobic known substance [68], possessing both brittleness and ductility [69], nontoxic, and inexpensive [70,71]. Table 3 shows a summary of the basic physical properties of graphene.

#### 2.2.2. Potential Applications of Graphene

The exceptional properties of graphene can be exploited in numerous applications, including biomedicine [82], membranes [83], sensors [84], energy harvesting and storage [85], composites and coatings [86], and functional devices [87], as shown in Figure 5. In addition, graphene is a promising material for the fabrication of smart and electronic textiles, where multiple functions can be combined in a single material. The large surface area and flexibility improve conformal contact, resulting in increased sensitivity [88]. Due to the atomic structure of carbon atoms in graphene, electrons can move at incredible speeds without scattering, saving energy that would otherwise be wasted in conventional conductors. The number of graphene layers and the coupling effects with the underlying substrate affect the electronic properties of the graphene system. Seamless integration of electronics into textiles can enable various applications, including flexible, stretchable, and foldable devices [89], electrodes, and electronic textiles that can be used in various fields.

## 3. Conductive Fabrics Based on Carbon Nanotubes

Conductive fabrics are usually made from various substrates, such as cotton [90], polyester [91], wool [92], and nylon [93], using numerous techniques, such as embroidery [94], knitting [95], spinning [96], coating [90], printing [97], dipping and drying, drop casting [98], and others. To make fabrics electrically conductive, there are usually two approaches: one approach is to incorporate conductive fillers, such as metal nanoparticles and carbon-based materials [99], graphene and carbon nanotubes, into the fabric. The second approach is to coat the fabric with a conductive polymer, such as PEDOT:PSS, which contains little or no metal. This review focuses on the fabrication of conductive fabrics from carbon-based materials.

### 3.1. Conductive Fabrics Based on SWCNTs 

Recently [100], conductive and flexible melt-blown fabrics were coated with SWCNTs by chemical vapor deposition, where the melt-blown fabrics were recycled from face masks. The results showed that the sheet resistance of the conductive fabrics depended on the deposition time and was 245, 116, and 57 Ω/□ for deposition times of 1 h, 2 h, and 3 h, respectively. It was also found that the sheet resistance decreased when the gold chloride dopant was used with values of 64, 54, and 26 Ω/□, respectively. Alamer et al. [101] fabricated a highly conductive cotton fabric impregnated with SWCNTs by using the filtration technique to produce conductive cotton. The advantage of this technique is that the residual solution that passes through the filter paper is collected in a beaker and stored for later use. This process was also safe, simple, and environmentally friendly, using renewable energy sources and using chemicals effectively. They found that the sheet resistance of the sample reached the minimum value of 0.006 Ω/□ at a concentration of 41.5 wt.%. They also ensured that the temperature behavior of the conductive cotton was consistent and reproducible for at least two months. Huang et al. [102] designed an electrode using a stretchable Lycra fabric, SWCNTs as conducting materials, and a dyeing and drying process. The impurities were first removed from the surface of the fabric using deionized water. Then, the fabric was stretched to 100% elongation and immersed in SWCNT ink, alcohol, and nitric acid, respectively. Then, the fabric was dried and stretched to allow the SWCNTs to penetrate the fabric and increase its conductivity. The resistance of the conductive electrode was stable after 5 × 10^2^ stretching cycles with a minimum sheet resistance of 65 Ω/□ at 35% tensile load. 

In another study, SWCNT ink was printed on a stretchable substrate using inkjet printing technique [103]. It was found that the sheet resistance of the conductive substrate depended on the number of coating layers and decreased as the number of layers increased. The minimum sheet resistance was 19.08 Ω/□ for the five-layer coating, and this value changed slightly after the sample was stretched. Zhang et al. [104] fabricated conductive cotton nylon with SWCNTs by immersion drying and then modified the fabric by plasma. The results showed that the sheet resistance of the modified fabric 2.0 k Ω/□ was lower than the sheet resistance of the unmodified fabric 4.9 k Ω/□, which was attributed to the increase in the surface roughness of the fabric. SWCNTs dispersed in dodecylbenzene sulfonic acid, sodium salt, were applied to polyester fabric by the coating dry cure method [105]. Before the coating process, the fabric was treated with plasma using different working gases and different treatment times. The results showed that this treatment led to an improvement in the antistatic properties of the polyester fabric. It was found that the antistatic property increased with increasing plasma treatment and then decreased. The effect of SWCNTs as absorbers of UV light for cotton fabrics was investigated in the study published by Mahmoudifard and Safi [106] and compared with ZnO and ${\mathrm{TiO}}_{2}$ absorbers. It was found that SWCNTs absorbed UV light with a high UPF value compared to ZnO and TiO2 absorbers. In another interesting study [107], a piezoresistive stretchable sensor based on SWCNTs and fabrics was fabricated, the joint movements of children were measured and compared with a rectangular sensor. It was found that the stretchable sensor had the same effect as the rectangular sensor with electrical resistance in the range of 280 Ω and 290 Ω. A flexible and stable supercapacitor with a high specific capacitance of about 70 to 80 Fg^−1^ was prepared by immersing cotton leaves in SWCNT ink [108]. The results showed that the sample exhibited high electrical conductivity with a sheet resistance of less than 1 Ω/□. In another study, SWCNTs dispersed in sodium dodecyl benzyl sulfonate and ethylene glycol were used to prepare conductive threads by the immersion drying method [109]. The results showed that the electrical conductivity was curiously dependent on the concentration of SWCNTs, with the resistance decreasing from 3.587 Ω to 0.01257 Ω as the concentration increased from 0.008049 wt% to 1.07269 wt%. In 2008, Shim et al. [110] fabricated a conductive cotton yarn by using SWNTs, MWNTs and polyelectrolytes and applying the immersion method. The cotton yarn becomes conductive after multiple immersions, with a resistivity as low as 20 Ω.cm^−1^. This approach is characterized by its speed, simplicity, robustness, low cost, and easy scalability. The fabrication of conductive Lycra fabric has also been investigated using the conductive materials SWCNTs and polyaniline, using the immersion drying process [111]. The results showed that the conductive fabric had a minimum sheet resistance of 35 Ω/□. This fabric was used to manufacture antenna which worked at 2.45 GHz with reflection coefficient of about ~18.6 dB. In another study [112], conductive cotton fabrics were prepared using a composite of SWCNTs and the conductive polymer PEDOT: PSS by the technique of drop casting. The effects of applying the composite in the cotton fabrics were studied, and the results showed that a cotton fabric composed of one layer of PEDOT: PSS between two layers of SWCNTs was electrically stable for four months, with a minimum sheet resistance of 0.006 Ω at a concentration of 41.5 wt.%. The metallic conductive threads were also prepared from SWCNTs, PEDOT:PSS, and a mixture of both [113]. The results showed that the electrical resistances depend on the fabrication process. The lowest sheet resistance was obtained for the sample prepared from a mixture of SWCNTs and PEDOT:PSS with a value of 0.0072 Ω, and a lower amount of composite of 1.729 mg. Table 4 show summary list of SWCNTs-based materials with details of their manufacturing processes and electrical properties.

### 3.2. Conductive Fabrics Based on MWCNTs

As discussed in the previous section, conductive fabrics made of SWCNTs have excellent electrical properties; however, SWCNTs are expensive, purification is difficult, and dispersion in liquid is also difficult. Therefore, many researchers focused on fabricating conductive fabrics using MWCNTs instead of SWCNTs because they are cheaper, can be produced in large quantities, and are more stable compared to SWCNTs. Rahman et al. [113] fabricated conductive and thermal cotton fabrics with MWCNTs using the dip and dry method. The results showed that the electrical conductivity of the conductive cotton was about 0.20 S m^−1^ with a sheet resistance of 1.67 kΩ/□ after four times of immersion. In addition, the thermal conductivity of the fabric was also increased by 70%. MWCNTs dispersed in DMF were used in the recent study presented by Alamer et al. [114] to prepare conductive cotton fabrics using the drop-casting and drying method. The sheet resistance of the conductive cotton was proportional to the MWCNT loading of the fabric and reached a value of 15.92 Ω/□ at a saturation concentration of 42.20 wt.%. Moreover, the conductive fabrics exhibited semiconductor behavior as the resistance decreased with increasing temperature. The conductive cotton fabrics were also prepared by immersing the fabrics in a dispersion of MWCNTs in sodium dodecyl sulphate [115]. The amount of MWCNTs was increased up to 20 times by repeating the immersion process, and the cotton fabrics with high MWCNT concentration exhibited a minimum sheet resistance of 2.5 kΩ.cm^−2^. The results also showed that the conductive cotton treated with HNO_3_ resulted in a reduction of sheet resistance to 1.5 kΩ.cm^−2^ which was attributed to the interaction between MWCNTs and cellulose through glycosidic bonds.

The conventional dyeing method was used to deposit synthetic MWCNTs on the surface of cotton fabrics [116]. The deposition of MWCNTs was uniform and permanent, and the results showed that the sheet resistance changed in the range of 5486 MΩ/□ to 0.433 MΩ/□ due to the change of the amount of MWCNTs from 100 mg to 500 mg. In addition, the mechanical properties of the conductive fabric were also improved, and the strength was increased by increasing the amount of MWCNTs, which was attributed to the effect of van der walls force between the nanotube particles and the cotton surface. In another study [117], MWCNTs were first dispersed by grafting dimethyl phosphite and perfluorohexyl iodine, then applied to cotton fabric by the impregnation-drying method. The conductive fabric had a sheet resistance of 225.6 kΩ/□ and exhibited UV resistance, with the UPF value reaching the maximum value of 121. Costa et al. [118] fabricated electrodes for supercapacitors based on cotton fabric and MWCNTs dispersed in sodium dodecylbenzene sulfonate. The MWCNT dispersion was applied to the surface of the conductive cotton using the dip-pad drying method. By repeating this method eight times, the resistance of the fabric electrodes reached the minimum value of 2.62 Ω.cm^−2^, had a specific capacitance of 8.01 F g^−1^, a high energy density of 6.30 Wh kg^−1^, and a cyclability of 5000. In the study presented by Nafeiea et al. [119], conductive wool fabrics were prepared using MWCNTs and carboxylated MWCNTs, both dispersed in water, and the effect of sodium dodecyl sulfate as an anionic surfactant and cetyltrimethylammonium bromide as a cationic surfactant was investigated. The results showed that the use of a cationic surfactant improved the dispersion of MWCNTs in water, while the dispersion of carboxylated MWCNTs in water was better without the use of a surfactant. The electrical conductivity of the wool fabric prepared with 5 g/L carboxylated MWCNTs reached a maximum value of 2 × 10^−3^ S cm^−1^, which is ten times higher than the conductivity of the wool fabric treated with MWCNTs. In the study presented by Kowalczyk et al. [120], MWCNTs were also dispersed in sodium dodecyl sulfate, then applied to polyester/cotton fabrics using the padding-drying method. The resistance depended on the number of pads and changed from 5.79 kΩ to 1.07 kΩ when the number of pads was increased from one to three, which was attributed to the formation of the MWCNT networks. Polyester fabric treated with MWCNTs dispersed in enzymes was used as an electrode in dye-sensitised solar cells [121], where the MWCNT dispersion was applied to the surface of the fabric by the tape-casting method. The sheet resistance of the treated fabric depended on the thickness of the coating and changed from 38 Ω/□ to 12 Ω/□ when the coating thickness increased from 5 µm to 28 µm. It was also found that the sheet resistance depended on the size of MWCNTs. The energy conversion efficiency of the conductive electrode reached about 5.69%. Hao et al. [122] fabricated flexible conductive cotton electrodes for supercapacitors using carboxyl MWCNTs. The carboxyl MWCNTs were deposited on the cotton fabric at high temperature and pressure by immersion method. The electrical resistance of the composite reached a value of 2.606 Ω with a high specific capacitance of 94.3 F g^−1^, and the sample exhibited good stability up to 3000 cycles. The conductive yarns based on MWCNTs were fabricated in the study presented by Abbas et al. [123,124], in which the spin-dry method was used for the fabrication process. The results showed that the resistance of the conductive yarn depended on the diameter of the yarn. It was 2.55 k Ω and 120 Ω for the yarns with diameters of 12 µm and 100 µm, respectively. In addition, the absorption coefficients of the conductive yarns were measured in the range of 50 MHz to 20 GHz and were found to depend on the diameter of the conductive yarns. Table 5 shows summary list of MWCNTs-based materials with details of their manufacturing processes and electrical properties.

### 3.3. Conductive Fabrics Based on Graphene

Incorporating graphene into textiles not only imparts conductivity to the textiles, but also enables the production of multifunctional textiles due to the excellent physical properties of graphene, as we discussed in Section 2. Gan et al. [125] fabricated conductive cotton fabrics using graphene nanoribbons by wet coating method. The mechanical and electrical properties of the fabrics were improved after repeating the wet coating method. The achieved low resistance was about 80 Ω with an increase in tensile stress and elastic modulus of 58.9% and 64.1%, respectively. In another study [126], the trapping method was used to fabricate conductive PET graphene-based fabrics. The main feature of this method is to reduce the insolubility of graphene so that it can easily penetrate the fabrics. The sheet resistance of the fabrics was strongly dependent on the graphene loading and changed from 77.9 MΩ/□ to 2.5 kΩ/□ when the graphene loading increased from 2.5 wt.% to 10.7 wt.%. Sahito et al. [127] developed a flexible and conductive cotton fabric coated with graphene nanosheets. Briefly, the charge of the surface of the cotton fabric was modified by cationization, which resulted in a positive charge that enabled strong bonding between the graphene oxide nanosheets and the cotton fabric and formed a uniform layer on the surface of the fabric, then the chemical reduction method was used to convert the graphene oxide nanosheets into graphene nanosheets. This conductive flexible cotton fabric with sheet resistance of 7 Ω/□ was used as a counter electrode for a dye-sensitive solar cell, and the calculated photovoltaic conversion efficiency was 6.93%. Ren et al. [128] fabricated conductive cotton fabrics with graphene oxide, where the graphene oxide was synthesized from graphite flakes, dispersed in DI water, applied to the cotton fabrics by a vacuum filtration method, and then reduced by a hot-pressing method. The sheet resistance was about 0.9 kΩ/□ and increased to about 1.2 kΩ/□ after 10 washing cycles. This conductive cotton fabric was used as a strain sensor and showed good stability up to 400 bending cycles. In an interesting method, Atta et al. [129] immersed cotton yarns in a graphene oxide dispersion, then reduced them with gamma rays. The resulting cotton yarns were used as portable supercapacitors and the specific capacitance reached a maximum value of 97 F/g. It was also found that the series resistance and charge transfer resistance depended on the graphene oxide concentration and reached a minimum value of 34 Ω and 22 Ω for the series resistance and charge transfer resistance, respectively. Maneval et al. [130] prepared conductive cotton yarns by using two methods: cationization to improve electrostatic interactions, and dip coating to coat the surface of cotton yarns with a graphene dispersion (see Figure 6). Before the yarn breaks, the electrical conductivity of the yarn reached a maximum of 1.1 S cm^−1^ at a graphene concentration of 14% by weight and under continuous mechanical stress.

Lu et al. [131] fabricated silk fabric with a high conductivity of a single fiber of 3595 S m^−1^ by using graphene oxide nanosheets as a conductive material and a coating reduction method. Briefly, the untreated silk fabric was immersed in bovine serum albumin, which generates a positive charge on the surface of the fabric and increases the absorption of the conductive material when the fabric is immersed in the graphene oxide nanosheet solution. Then, a hydrazine vapor reduction method was used to reduce the graphene oxide on the fabric. In the study presented by Zulan et al. [132], the conductive silk fabric was also prepared with graphene oxide after the fabric was modified. The modification of the silk fabric was performed as follows: the fabric was immersed in a solution containing regenerated silk fibroin as an electrostatic adhesive, deionized water, and bovine serum albumin. The modified fabric was coated with graphene oxide, then thermally reduced to convert the graphene oxide into graphene. The results showed that the conductive silk fabric was thermally stable and exhibited an electrical conductivity of $3.06\times 10^{-6}$ S cm^−1^. In another study [133], a flexible, stable, conductive cotton yarn with an electrical conductivity of about 1.0 S cm^−1^ was prepared using reduced graphene oxide and a dip coating and reduction method. The results also showed that the conductive cotton yarn exhibited mechanical stability up to 1000 cycles and absorbed UV irradiation of about 1.0 mA/W under bending deformation. Yarns from Calotropis gigantean [134], which have a unique structure, excellent hydrophilicity, and lower natural longitudinal crimp, were used to produce conductive yarns on a large scale by dyeing graphene oxide onto the surface of the yarn and applying a reduction process (see Figure 7). The obtained conductivity of the treated yarn depended on the concentration of graphene oxide and reached a maximum value of 6.9 S m^−1^ at high concentration and was shown to be resistant to washing, which was due to the hydrogen bonding formed between the fiber and graphene during the dyeing process.

Molina et al. [135] fabricated conductive fabrics by chemical reduction of graphene oxide on polyester fabric. The resistance of the fabric decreased from $10^{11}$ Ω. cm^2^ for the untreated fabric to 23.15 Ω. cm^2^ for the fabric coated with three layers of reduced graphene oxide. In another study [136], the knitted fabric was also immersed in graphene oxide solution, then subjected to a reduction process. The sheet resistance of the resulting promoted fabric was dependent on the amount of reduced graphene oxide in the fabric and the number of immersion cycles. It reached the value of 0.19 MΩ/□ after 15 dipping cycles. The graphene/polyurethane composite material and the dip coating method [137] were used to fabricate a conductive para-aramid fabric, as shown in Figure 8. It was found that the sheet resistance and electrical capacitance of the conductive fabric decreased as the number of dip coating cycles increased due to the increase in the amount of composite materials. The minimum sheet resistance and electrical capacitance of the conductive fabric reached 75 kΩ/□  and 89.4 pF, respectively, after 5 dip coating cycles. It was also found that this sample can be used for a heat-resistant para-aramid knitted glove with a phone touch screen when hot-pressed at 140 degrees.

A conductive stretch-sensitive fabric was fabricated using graphene oxide nanosheets and a reducing deposition method [138]. Briefly, graphene oxide nanosheets were deposited on nylon/polyurethane fabric, then reduced with sodium borohydride. The results showed that the electrical resistivity of the conductive fabric with a value of 112 ${k\Omega \;m}^{-2}\;$ was four times lower than that of the untreated nylon/polyurethane fabric. In addition, the electrical resistivity of the conductive fabric increased from 112 ${k\Omega \;m}^{-2}\;$ to 154 ${k\Omega \;m}^{-2}$ after eight washes. The conductive fabric was also used to fabricate a strain sensor in the strain range of 0 to 30 percent, and the strain sensor exhibited good sensitivity and stability. Ba et al. [139] found a method to improve the bonding between the graphene and the functional group on the cotton fabric using karaya gum as a bioinspired exfoliating agent, in which the synthesized graphene solution was applied to the surface of the cotton fabric by dip coating or brush coating. The electrical conductivity of the conductive cotton fabric reached a maximum value of 13,000 S m^−1^ at a graphene concentration of 6 wt.%. Another interesting method to improve the bonding between graphene and cotton fabric and to fabricate a scalable conductive fabric with a length of 150 m and a sheet resistance of about 11.9 Ω/□ was presented in the study by Afroj et al. [140]. The conductive graphene dispersion was prepared using the microfluidization technique for natural graphite flakes, then applied to the cotton fabric using the pad dry curing method. Another important observation was that the conductivity of the conductive cotton did not change even after washing ten times. The screen printing method [141] was used to print graphene ink on the surface of the textile after the textile was modified using heat transfer technology. The sheet resistance of the conductive textile reached a minimum value of 100 Ω/□ after three printing cycles, and this textile was used to fabricate a conductive electrode for electrocardiogram monitoring. It was found that the efficiency of the graphene electrode was comparable to the conventional electrode. In another study, the screen printing method with graphene ink was also used, but for the two sides of the cotton fabric in the study presented by Zhang et al. [142], and the fabric produced was used as a portable heater (see Figure 9). The small voltage difference of 3 V applied to the conductive fabric resulted in a high heating temperature, 52.6 °C, which confirmed that this conductive fabric could be used as a wearable heater. The conductive cotton fabric also exhibited a high electrical conductivity of 1.18 × 10^4^ S m^−1^. The biocompatible conductive fabric sensor was fabricated using graphene nanoplatelets dispersed in a water-based ink and screen-printed onto the fabric surface [143]. The results showed that the fabric sensor was stable and sensitive, that the stiffness of the fabric increased with the amount of material applied, and that the electrical conductivity reached the maximum value of about 10.26 S m^−1^ at a graphene concentration of 3.8 wt%.

In addition, Yapici et al. [144] also fabricated electrocardiogram electrodes based on nylon fabric coated with reduced graphene oxide using the immersion drying method. The resulting electrode had an electrical conductivity of 4.5 S cm^−1^, and this value was stable up to five washing cycles. In the study by Sahito et al. [145], the surface of a cotton fabric was modified with bovine serum albumin, which resulted in a positive charge on the surface. In their study, the electrical properties of cotton and modified cotton were compared after they were immersed in graphene oxide. The results showed that the amount of graphene oxide in the modified cotton was greater than the amount of graphene oxide in the cotton. Then, the graphene oxide was reduced and converted to graphene by the chemical vapor reduction method. The minimum sheet resistances obtained were 40 Ω/□ and 510 Ω/□ for the conductive cotton and the conductive modified cotton, respectively. The conductive nylon 6 fabric was prepared by depositing reduced graphene oxide on the surface of the fabric, which was presented in the study by Yun et al. [146]. A new method was used in this process: electrostatic self-assembly of graphene oxide and bovine serum albumin to improve the adhesion of graphene sheets on the fabric was used to deposit them on the surface of the fabric, followed by a low-temperature reduction process. The conductive fabric exhibited high electrical conductivity of 1000 S cm^−1^, which is not affected by bending and washing cycles. In another process [147], UV light was used to reduce graphene oxide on cotton and wool fabrics without using a reducing agent or high annealing temperature. Briefly, graphene oxide was first applied to the surface of the fabric using the brush coating drying method, the process was repeated to increase the concentration of the materials, then the fabric was irradiated with UV light to reduce the graphene oxide and convert it to graphene. It was found that the sheet resistance reached a minimum value of 100.80 kΩ/□ and 45 kΩ/□ when the conductive cotton and wool fabrics were irradiated with UV light, respectively. Table 6 shows summary list of graphene-based materials with details of their manufacturing processes and electrical properties.

## 4. Summary

This review article summarizes the method of designing and fabricating electrical, flexible, and lightweight conductive fabrics with embedded SWCNTs, MWCNTs, and graphene, and their applications in the field of smart textiles. The development of smart textiles was carried out in three stages: the first stage is to impart conductivity to the textiles, the second stage is to fabricate the smart textiles, and the final stage is to functionalize the conductive yarns. Carbon-based materials, particularly SWCNTs, MWCNTs, and graphene, are discussed from their structure, physical properties, and potential applications to their use in the design and fabrication of conductive fabrics with a wide range of electrical conductivity and interesting physical properties that make them suitable for various wearable electronic applications. We come to the following conclusions:Conductive fabrics based on SWCNTs have been prepared by various fabrication methods: chemical vapor deposition, filtration technique, dyeing and drying method, inkjet printing method, dipping and drying method, and drop-casting method. It has been shown that the electrical conductivity of conductive fabrics and sheet resistance has a wide range from low to high values, and depends on various factors, such as: deposition times, dopants, content of SWCNTs in the fabric, stretching cycles, number of coating layers, treatment of the fabric with plasma, and mixing of SWCNTs with other carbon-based materials, such as MWCNTS and graphene, or with conductive polymers, such as polyaniline and PEDOT:PSS. These SWCNTs based fabrics have been used in various applications such as: UV light shielding, piezoresistive sensor, supercapacitor, antenna, and metal thread.Conductive fabrics based on MWCNTs were prepared by various fabrication methods: Dipping and drying, drop casting and drying, dipping, impregnation and drying, dipping and drying, and tape casting. The fabricated conductive fabrics exhibited a wide range of electrical conductivity, which was influenced by several factors: Size of MWCNTs, number of dipping operations, type of organic solvents, content of MWCNTs, temperature, dopants, repetition of the fabrication process, use of anionic and cationic surfactants, and use of enzymes. These conductive fabrics based on MWCNTs have been used as electrodes in supercapacitors, and as electrodes in dye-sensitised solar cells and block UV light.Conductive graphene-based fabrics have been fabricated using graphene, graphene nanoribbons, graphene oxide, graphene nanosheets, natural graphite flakes, and graphene ink. Various methods were used in the fabrication process: wet coating, trapping method, chemical reduction method, vacuum filtration—hot press method, dipping and gamma ray reduction method, cationization—dip coating, coating and reduction method, dyeing and reduction method, dipping and reduction method, reducing deposition method, brush coating, pad dry curing method, screen printing method, chemical vapor reduction method, electrostatic self-assembly—low temperature reduction process, UV reduction method. The electrical properties of graphene-based materials are influenced by several factors: repetition of the manufacturing process, graphene content, graphene oxide concentration, tensile stress, cationization process, number of coating layers, number of dipping cycles, exfoliation agent, and number of printing cycles. Thus, the conductive fabrics produced have been used in various applications: electrode for a dye-sensitive solar cell, strain sensor, portable supercapacitors, UV blockers, heat-resistant gloves, electrocardiogram electrodes, and portable heaters. Finally, the field of smart textile fabrication from carbon-based materials is developing rapidly but needs further development to bring these applications from small scales (in research laboratories) to large scales (industrial applications). Therefore, more basic research is needed to enable the next wave of smart textile products.

## Figures and Tables

**Figure 1 polymers-14-05376-f001:**
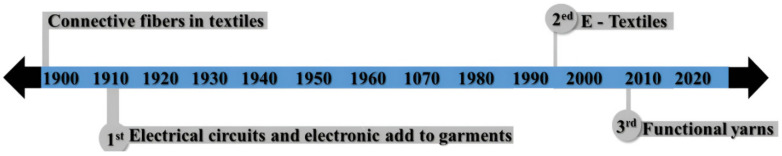
This timeline shows different generations of electronic textiles. Adapted [7].

**Figure 2 polymers-14-05376-f002:**
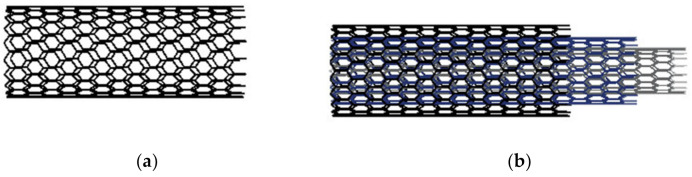
Structure of carbon nanotubes (**a**) SWNTs and (**b**) MWNTs.

**Figure 3 polymers-14-05376-f003:**
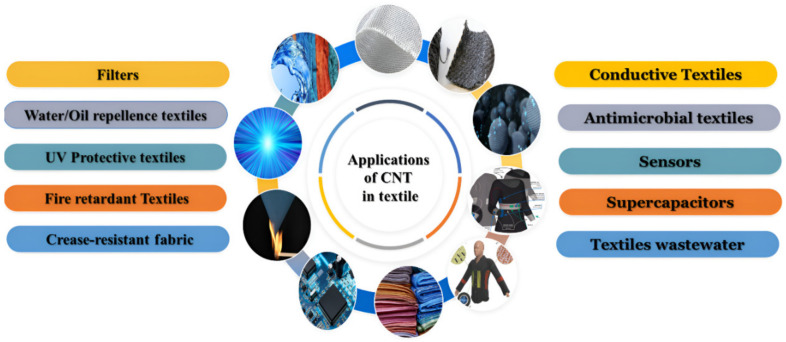
Potential applications for the use of carbon nanotubes in textiles.

**Figure 4 polymers-14-05376-f004:**
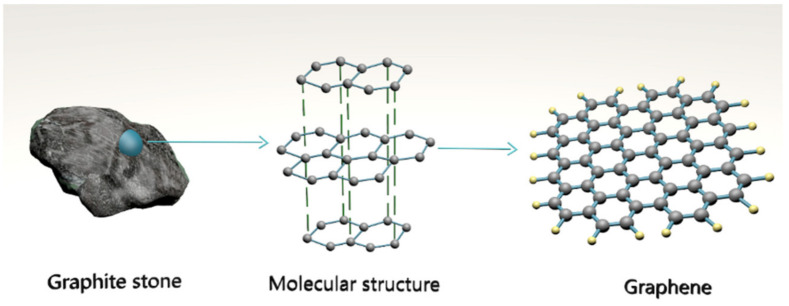
Structure of graphene as a honeycomb lattice of carbon atoms.

**Figure 5 polymers-14-05376-f005:**
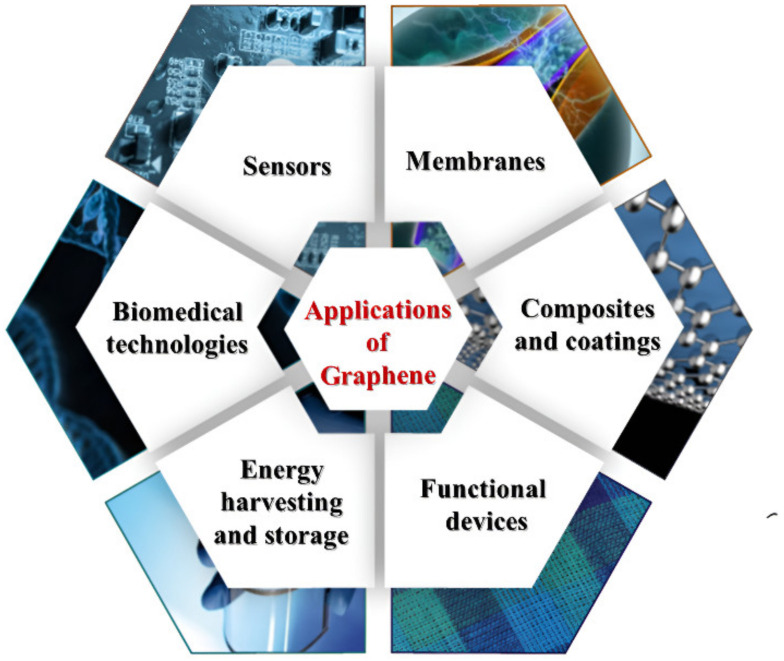
Potential Applications of graphene.

**Figure 6 polymers-14-05376-f006:**
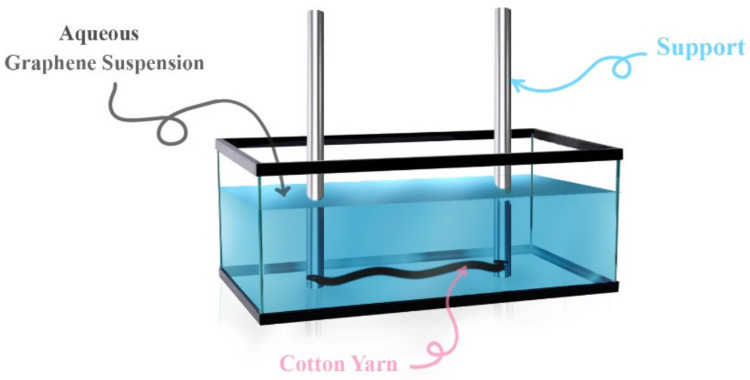
Schematic representation of the device used for the coating of cotton yarns with graphene sheets [130].

**Figure 7 polymers-14-05376-f007:**
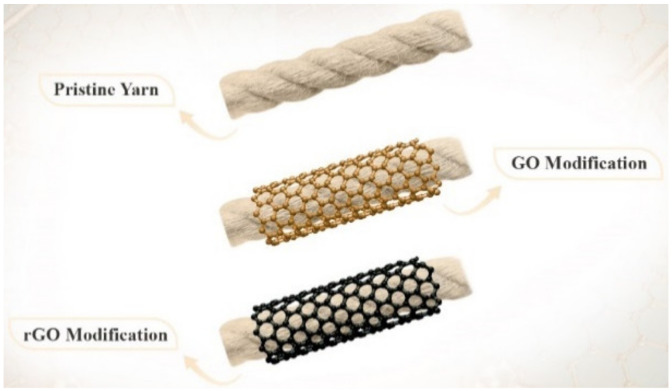
Images of the pristine, GO-modified, and rGO-modified CGYs [134].

**Figure 8 polymers-14-05376-f008:**
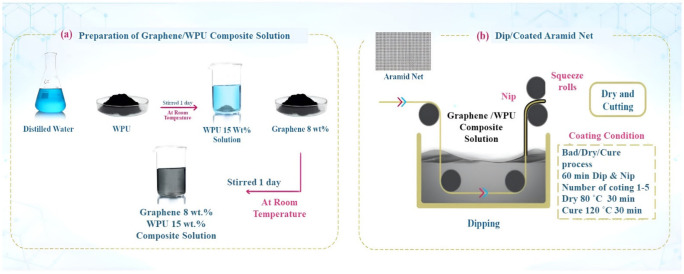
Illustration of the fabrication process for graphene/WPU with dip coating on para-aramid knitted fabric, which consists of two steps: (**a**) preparation of graphene/WPU composite solution, (**b**) dip coating of para-aramid knitted fabric with different coating cycles [137].

**Figure 9 polymers-14-05376-f009:**
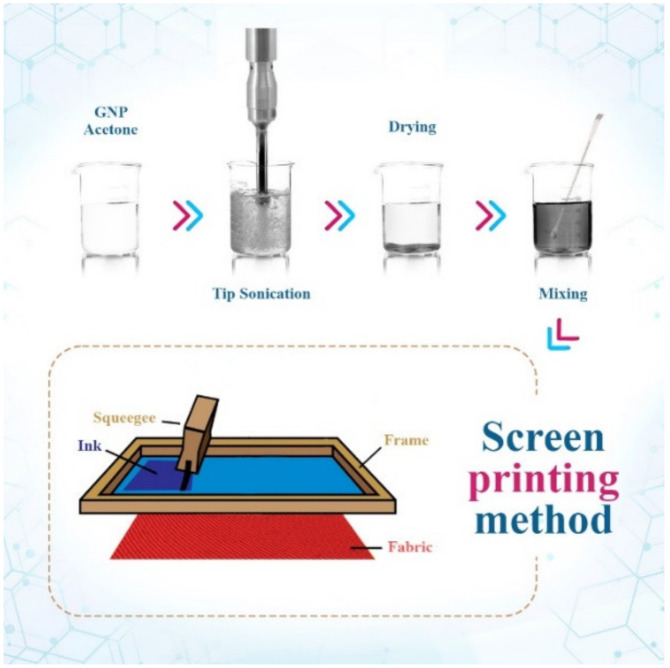
Screen printing process for the production of conductive cotton with graphene ink [143].

**Table 1 polymers-14-05376-t001:** Comparison between SWNTs and MWNTs.

SWCNTs	MWCNTs	References
Single graphene layer	Multiple graphene layers	[40,44]
SWNTs have a diameter of 0.4 to 3.0 nm and a length of 20 to 1000 nm.	The outer diameters are between 2 and 100 nm, the inner diameters between 1–3 nm and the lengths between 1 and 50 m.	[29,45]
The synthesis of SWCNTs requires the use of a catalyst.	The synthesis of MWCNTs can in fact be made no need for a catalytic	[40,44]
Bulk production is challenging because it requires precise control of growth and environmental conditions.	Bulk production is simple.	
Purity is poor	Purity is high	
Less deposits in the body	More deposits in the body	
It is more flexible and can be twisted effortlessly.	It is complicated to twist.	
Characterization and evaluation are simple	It has a very complex structure	

**Table 2 polymers-14-05376-t002:** Summary of the main properties of SWCNTs and MWCNTs.

Properties	Unit	SWCNTs	MWCNTs	References
Specific gravity	$g/m^{3}$	0.8–1.3	1.8–2.6	[46]
Resistivity	μΩ/cm	5–50	5–50	
Young’s modulus	TPa	∼1	∼1–0.3	[41,42]
Thermal conductivity	$W.m^{-1}K^{-1}$	3000–6000	2000–3000	[41,43]
Electrical conductivity	S/m	$10^{2}–10^{6}$	$10^{3}–10^{5}$	[43]
Thermal stability in air	°C	550–650	550–650	
Specific area	$m^{2}/g$	400–900	200–400	

**Table 3 polymers-14-05376-t003:** Summary of the basic physical properties of graphene.

Quantity	Values	References
Tensile strength	130 GPa	[72]
Young’s modulus	1TPa	[73]
Weight	0.77 mg/$m^{2}$	[74]
Thermal conductivity	~3000–5000 W m^−1^ K^−1^	[75,76]
Mobility of charge carrier	2 × 10^5^ cm^2^ V^−1^ s^−1^	[77,78,79]
Electrical conductivity	~$3.6\times 10^{8}\;S/m$	[80]
Transmittance	≈97.7%	[81]

**Table 4 polymers-14-05376-t004:** Shows summary list of SWCNTs-based materials with details of their manufacturing processes and electrical properties.

Year	Substrate	Coated with	Method	Electrical Properties	References
2022	Melt blown fabrics	SWCNTs	Chemical vapor deposition	57 Ω/□	[100]
		SWCNTs + gold chloride		26 Ω/□	
2020	Cotton	SWNTs	Filtration	0.006 Ω/□	[101]
2017	Lycra	SWCNTs	Dyeing drying	65 Ω/□	[102]
2014	Thin films	SWCNTs	Inkjet printed	19.08 Ω/□	[103]
2012	Cotton nylon	SWCNTs	Dyeing drying	2.0 k Ω/□	[104]
2015	Polyester	SWCNTs	Coating-dry-cure	-	[105]
2012	Cotton	SWCNTs + ZnO + ${\mathrm{TiO}}_{2}$	Dyeing drying	-	[106]
2020	Polyester spandex	SWCNTs + polyurethane	Dry curing machine	280–290 Ω	[107]
2010	Cotton	SWCNT ink	Dyeing drying	1 Ω/□	[108]
2020	Cotton thread	SWCNTs + SDBS	Dip coating	0.01257 Ω	[109]
2008	Cotton yarn	SWNTs + MWNTs + polyelectrolyte	Dip coating	20 Ω/cm	[110]
2017	Lycra	SWCNTs	Dyeing drying	35 Ω/□	[111]
2021	Cotton thread	SWNTs + PEDOT: PSS	drop-casting	0.0072 Ω	[112]

**Table 5 polymers-14-05376-t005:** Summary list of MWCNTs-based materials with details of their manufacturing processes and electrical properties.

Year	Substrate	Coated with	Method	Electrical Properties	References
2015	Cotton	MWCNTs	Dyeing drying	1.67 kΩ/□ 0.20 S m^−1^	[115]
2022	Cotton	MWCNTs	Dyeing drying	15.92 Ω/□	[116]
2015	Cotton	MWCNTs	Dyeing drying	1.5 ${\Omega k\;\mathrm{cm}}^{-2}$	[117]
2019	Cotton	MWCNTs	Dyeing	0.433 MΩ/□	[118]
2020	Cotton	MWCNTs	Dipping drying	225.6 kΩ/□	[119]
2020	Cotton	MWCNTs	Dip-pad-dry	2.625 Ω ${\mathrm{cm}}^{-2}$	[120]
2016	Wool	MWCNTs	-	2 × $10{\;}^{-3}\;$ S cm^−1^	[121]
2015	Polyester/Cotton	MWCNTs	Padding machine	1.03 × $10{\;}^{3}$ Ω/□	[122]
2015	Polyester	MWCNTs	Tape casting	15 Ω/□	[123]
2018	Cotton	MWCNTs-COOH	Dyeing drying	2.606 Ω	[124]

**Table 6 polymers-14-05376-t006:** Summary list of graphene-based materials with details of their manufacturing processes and electrical properties.

Year	Substrate	Coated with	Method	Electrical Properties	References
2015	Cotton	Graphene nanoribbon	Wet coating	80 Ω	[125]
2014	Cotton yarn	Graphene/graphite	Trapping	2.5 kΩ/□	[126]
2016	Cotton	Graphene nanosheets	Dyeing drying	7 Ω	[127]
2017	Cotton	Graphene oxide	Vacuum filtration	0.9 kΩ/□	[128]
2021	Cotton	Graphene oxide	Immersing	22–34Ω	[129]
2021	Cotton	Graphene	Dyeing drying	1.1 S cm^−1^	[130]
2015	Silk	Graphene oxide	Immersing	3595 S m^−1^	[131]
2019	Silk	Graphene Oxide	Dip-pad	3.06 × $10{\;}^{-4}$ S cm^−1^	[132]
2019	Cotton thread	Graphene oxide	Dip-coating chemical reduction	~1.0 S cm^−1^	[133]
2022	Gigantea yarn	Graphene oxide	Pad dyeing	6.9 S m^−1^	[134]
2013	Polyester	Graphene oxide	Chemical reduction	23.15 Ω. cm^2^	[135]
2017	knitted	Graphene oxide	Dyeing drying	0.19 MΩ/□	[136]
2019	Para-aramid	Graphene/waterborne/polyurethane	Dip coating	7.5 × $10{\;}^{4}$ Ω/□	[137]
2017	Nylon	Graphene oxide	Dip coating	112 KΩ$/m^{2}$	[138]
2020	Cotton	Graphene	Dip- and brush-coated	13000 S m^−1^	[139]
2020	Poly-cotton	Graphene	Pad dry	11.9 Ω/□	[140]
2020	Cotton	Graphene	Screen-printing	100 Ω/□	[141]
2021	Cotton	Graphene ink	Screen-printing	1.18 × $10{\;}^{4}$ S m^−1^	[142]
2021	Polyester elastane	Graphene nanoplatelets	Screen-printing	10.26 S m^−1^	[143]
2015	Nylon	Graphene oxide	Dip coating	4.5 S cm^−1^	[144]
2015	Cotton	Graphene oxide	Dip coating	40 Ω/□	[145]
	Modified cotton			510 Ω/□	
2013	Nylon yarns	Graphene oxide	Electrostatic self-assembly and low temperature reduction	1000 S/m	[146]
2014	Cotton	Graphene oxide	Brush coating drying	100.8 kΩ/□	[147]
	Wool			45 kΩ/□	

## Data Availability

Not applicable.

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
