# Peer review of "Fabrication of Conductive Fabrics Based on SWCNTs, MWCNTs and Graphene and Their Applications: A Review"

_polymers, 2022, doi:10.3390/polym14245376_

Round 1

Reviewer 1 Report

Comments to the Author

Title: Fabrication of smart textiles based on SWCNTs, MWCNTs and graphene and their applications: A review

The authors should apply the following comments. So, the manuscript can be accepted after major revision.

Sections 2.1.1 and 2.1.2 provide general information on carbon nanotubes. This part is not relevant to the topic of the review. Therefore, only a few sentences should be written in this section. In section 2.1.3, only the last paragraph refers to the overview topic. So part 2.1 should be rewritten.

Section 2.2 also has many off-topic parts. Therefore, this part should be rewritten.

In general, section 2 does not live up to its name.

Section 3 is considered the main part of the article. This section is devoted to the preparation of conductive textile.

Thus, the title of the article and the text do not exactly match each other. The article is not devoted to smart textile, but to conductive fabrics, which are a type of them.

Reviewer 2 Report

Author have presented a review article on fabrication of smart textiles based on SWCNTs, MWCNTs and graphene and their applications. The review article is a good contribution and ill be useful. Following suggestions will be useful to further improve the manuscript.

·        Figures needs to be improved, as some of the figures are smaller in size and text is not easily readable. Increasing the size of figures/text will be better for readability.

·        Some places have very large fonts for values e.g. first line of section 2.2.3.

·        At some places, units are not properly displayed, this could potentially happen during conversion, please carefully check the manuscript. e.g. section 3.1 Ohm/?? and Table 5.

·        The article discusses applications of smart textiles based on SWCNTs, MWCNTs and graphene in wearable electronics, medical, healthcare, military etc. Most of the work presented is based on DC, whereas RF based work is limited that supports wireless communication, such as reference [111] in this work. It will be worthwhile to add some comments related to RF based work. Below work could be useful in this regard.

https://ieeexplore.ieee.org/document/6599003

https://doi.org/10.1016/j.xcrp.2022.100989

https://eudl.eu/doi/10.4108/eai.28-9-2015.2261421

10.1007/s40820-021-00721-4

·        Please double check the manuscript for typos and grammatical errors.

Author Response

please check the attchment 

Round 2

Reviewer 1 Report

Reviewer:

Comments to the Author

Title: Fabrication of smart textiles based on SWCNTs, MWCNTs and graphene and their applications: A review

The authors should apply the following comments. So, the manuscript can be accepted after minor revision.

The review article consists of 29 pages. 13 pages of the 21 page (main text) text are devoted to the preparation, properties and applications of conductive fabrics. Besides these, the summary part about conductive fabrics. Therefore, smart textiles of the name of the review raises questions.

Author Response

Thanks for your comment 

we updated the title of the manuscript to

Fabrication of conductive fabrics based on SWCNTs, MWCNTs and graphene and their applications: A review